# Pulmonary Toxicity of Polystyrene, Polypropylene, and Polyvinyl Chloride Microplastics in Mice

**DOI:** 10.3390/molecules27227926

**Published:** 2022-11-16

**Authors:** Isaac Kwabena Danso, Jong-Hwan Woo, Kyuhong Lee

**Affiliations:** 1Inhalation Toxicology Center for Airborne Risk Factor, Korea Institute of Toxicology, 30 Baekhak 1-gil, Jeongeup 56212, Jeollabuk-do, Republic of Korea; 2Department of Human and Environmental Toxicology, University of Science & Technology, Daejeon 34113, Republic of Korea; 3Biosafety Research Institute and Laboratory of Pathology, College of Veterinary Medicine, Jeonbuk National University, Iksan 54596, Jeollabuk-do, Republic of Korea

**Keywords:** polystyrene, polypropylene, polyvinyl chloride, C57BL/6 mice, BALB/c mice, ICR mice, NLRP3 inflammasome

## Abstract

Globally, plastics are used in various products. Concerns regarding the human body’s exposure to plastics and environmental pollution have increased with increased plastic use. Microplastics can be detected in the atmosphere, leading to potential human health risks through inhalation; however, the toxic effects of microplastic inhalation are poorly understood. In this study, we examined the pulmonary toxicity of polystyrene (PS), polypropylene (PP), and polyvinyl chloride (PVC) in C57BL/6, BALB/c, and ICR mice strains. Mice were intratracheally instilled with 5 mg/kg of PS, PP, or PVC daily for two weeks. PS stimulation increased inflammatory cells in the bronchoalveolar lavage fluid (BALF) of C57BL/6 and ICR mice. Histopathological analysis of PS-instilled C57BL/6 and PP-instilled ICR mice showed inflammatory cell infiltration. PS increased the NLR family pyrin domain containing 3 (NLRP3) inflammasome components in the lung tissue of C57BL/6 and ICR mice, while PS-instilled BALB/c mice remained unchanged. PS stimulation increased inflammatory cytokines, including IL-1β and IL-6, in BALF of C57BL/6 mice. PP-instilled ICR mice showed increased NLRP3, ASC, and Caspase-1 in the lung tissue compared to the control groups and increased IL-1β levels in BALF. These results could provide baseline data for understanding the pulmonary toxicity of microplastic inhalation.

## 1. Introduction

With advancements in human society, plastics have been used in many products and materials in daily life owing to their low cost, easy production, and extreme versatility. An estimated 8.3 billion metric tons of plastics have been produced since the beginning of their mass production in the 1950s, with an annual production of approximately 370 million metric tons in 2019 [1,2]. The increase in plastic use has concomitantly increased environmental pollution and with it, exposure predominantly to the marine- and human system [3,4,5]. The discarded plastics are disintegrated into smaller pieces by physical forces such as ultraviolet radiation [6]. Previous studies have evaluated the toxicological effects of microplastics on marine life, hypothesizing adverse exposure in humans resulting from the consumption of the exposed marine food. It has also been reported that microplastics present in the ocean are present in the atmosphere as well [5,7]. These atmospheric microplastics float from various regions, including roads, sea spray, agricultural dust, and dust from population centers [7,8]. Studies have reported the detection of airborne microplastics in lung tissues, they enter the human respiratory system through inhalation and lead to respiratory injuries due to their accumulation in the lungs [9,10,11,12]. The various microplastics detected in human lung tissues include polystyrene (PS), polypropylene (PP), and polyvinyl chloride (PVC) [9,12]. These results suggest that toxicity assessment of various microplastics in the respiratory system is required since the toxic effects of microplastics in the respiratory system remain poorly understood.

Toxicity assessments for chemicals are performed in animals due to ethical concerns; therefore, it is necessary to identify animal strains sensitive to chemical exposure. A common hypothesis is that, animals and humans exposed to chemicals have similar toxicity responses [13]. Previous studies have indicated different immune responses to various chemicals in their toxicity assessments of chemical exposure to the respiratory system of various mouse strains, including C57BL/6, BALB/c, and ICR mice [13,14,15]. In particular, these mouse strains had different genetic backgrounds that affected the immune response processes such as inflammation, cytokine production, and tissue remodeling to chemical exposure [16]. Based on these data, we hypothesized that genetic background might influence immune responses to microplastic-stimulated lung injury in different mouse strains.

The NLR family pyrin domain containing 3 (NLRP3) inflammasome is a large multiprotein complex that is responsible for the activation of inflammatory responses in the innate immune system [17,18,19]. NLRP3 inflammasome activation is initiated by various danger signaling pathways, such as pathogen-associated molecular patterns (PAMPs) or danger-associated molecular patterns (DAMPs) [18,19,20,21]. In addition, activation of the NLRP3 inflammasome is involved in the regulation of proteolytic cleavage, maturation, and secretion of the pro-inflammatory cytokines interleukin (IL)-18 and IL-1β [19,22]. Previous studies have reported that microplastic expo-sure causes physiological injuries to organs such as the heart, colon, and liver through activation of the NLRP3 inflammasome signaling pathway [23,24,25]. Interestingly, NLRP3 inflammasome activation by chemical, pathogen, and allergen exposure is a key factor that drives various pulmonary diseases, including asthma, pulmonary fibrosis, and chronic obstructive pulmonary disease (COPD) [26,27,28,29]. However, pulmonary injuries and the ensuing immune responses resulting from microplastic exposure remain unclear. 

In this study, we investigated the pulmonary toxicity response in three strains of mice (C57BL/6, BALB/c, and ICR) by analyzing the inflammatory cellular changes and histo-pathological analysis upon stimulation with three microplastics (PS, PP, and PVC). We also examined NLRP3 inflammasome activation in three strains of PS- and PP-instilled mice. 

## 2. Results

### 2.1. Changes in Inflammatory Cells after Exposure to PS, PP, and PVC in C57BL/6, BALB/c, and ICR Mouse Strains

To investigate the lung toxicity response to PS, PP, and PVC, we used three strains of mice instilled with the three microplastics. Our results showed that the number of total cells, macrophages, eosinophils, neutrophils, and lymphocytes in the bronchoalveolar lavage fluid (BALF) of PS-instilled C57BL/6 mice was significantly increased compared to those in the vehicle control (VC) group. However, the inflammatory cell numbers in PP- and PVC-stimulated C57BL/6 mice were unchanged (Figure 1A). In BALB/c mice, PS, PP, and PVC stimulation showed no changes in inflammatory cell numbers compared to those in the VC group (Figure 1B). Interestingly, the total number of cells and macrophages in the BALF of PS- and PP-instilled ICR mice were significantly increased compared to the control groups, while PP stimulation also increased the number of neutrophils. However, PVC-instilled ICR mice showed no changes in inflammatory cell counts (Figure 1C).

### 2.2. Histopathological Observations of the Lung Tissue of Mice Exposed to Microplastics 

In the present study, histopathological analysis involved determining severity scores of individuals according to the degree of severity of lung injury following intratracheal instillation of the three microplastics. Histopathological analysis of the lung tissue of PS-instilled C57BL/6 mice revealed inflammatory cell infiltration (Figure 2A, Table 1). Inflammatory cell infiltration was not observed in the lung tissue of PS-, PP-, and PVC-instilled BALB/c mice (Figure 2B, Table 2). PP stimulation in ICR mice induced inflammatory cell infiltration in the lung tissue compared with in the control groups. However, inflammatory cell infiltration in the lung tissue of PS- and PVC-instilled ICR mice was observed as a small lesion (Figure 2C, Table 3). 

### 2.3. PS Microplastic Stimulation Activates NLRP3 Inflammasome in C57BL/6 and ICR Mice 

The levels of NLRP3 inflammasome components in the lung tissue of PS-instilled C57BL/6 and ICR mice were significantly increased compared to that of the control groups, while NLRP3 protein levels in the lung tissue of PS-instilled ICR mice remained unchanged (Figure 3A–D). Additionally, the levels of inflammatory cytokines in the BALF, including IL-1β and IL-6, were significantly higher in the PS-instilled C57BL/6 mice than in the control groups (Figure 3E,F). In PS-stimulated ICR mice, IL-1β levels in BALF were increased significantly compared to those in the control groups, but IL-6 protein levels remained unchanged (Figure 3E,F). 

### 2.4. PP Microplastic Stimulation Activates NLRP3 Inflammasome in ICR Mice 

The levels of NLRP3 inflammasome components such as NLRP3, ASC, and Caspase-1 in the lung tissue of PP-instilled ICR mice were significantly increased compared to those of the control groups but not in C57BL/6 and BALB/c mice (Figure 4A–D). In addition, IL-1β levels in the BALF of PP-stimulated ICR mice were significantly increased compared to those in the control groups (Figure 4E). Interestingly, the IL-6 levels were not significantly different between the PP-instilled mice and the control groups (Figure 4F). 

## 3. Discussion

We investigated the differences in pulmonary toxicity responses to exposure to three microplastics (PS, PP, and PVC) in three mouse strains. Inflammatory cells, including macrophages, eosinophils, neutrophils, and lymphocytes, in the BALF of PS-instilled C57BL/6 mice were significantly increased compared to those in the control groups. Additionally, inflammatory cells, including total cells, macrophages, and neutrophils, were significantly increased in the BALF of PP-instilled ICR mice compared to those of the control groups. In contrast, PVC exposure had no impact on the inflammatory cellular changes in the three strains of mice. Interestingly, histopathological analysis of lung tissue in PS-instilled C57BL/6 and PP-instilled ICR mice showed inflammatory cell infiltration. However, no significant inflammatory responses were observed in BALB/c mice after stimulations with the three microplastics. We investigated the levels of inflammatory cytokines and NLRP3 inflammasome activation in response to PS and PP stimulation in the three mouse strains. NLRP3 inflammasome components, including NLRP3, ASC, and Caspase-1, in the lung tissue of PS-instilled C57BL/6 mice were significantly increased compared to those of the control groups. In addition, levels of inflammatory cytokines, such as IL-1β and IL-6, were increased in the BALF of PS-instilled C57BL/6 mice. Interestingly, NLRP3 inflammasome components and IL-1β levels were significantly increased in PP-instilled ICR mice compared with those in the control groups. These results suggest that microplastic stimulation contributes to lung inflammation via NLRP3 inflammasome activation.

Occupational exposure to airborne particulate matter such as microplastics, increases the risk of respiratory damage, impairs pulmonary functions, and increases serum interleukin levels; while chronic exposure has been reported to cause adverse pulmonary diseases such as fibrosis and cancer [11,30]. Previous research has suggested that workers in PVC industries could develop pneumoconiosis after 10 years of exposure to PVC, and that increase in airborne PVC concentration along with its long-term inhalation and exposure could result in fibrosis or carcinogenesis via chronic inflammatory stimulation. [10,30,31]. A study by Xu et al., reported that 50 mg/kg of PVC single intratracheal instillation in rats caused acute inflammatory responses in lung tissue [32]. In addition, oral administration of 100 mg/kg of PVC microplastics in mice caused intestinal barrier damage and dysfunction including metabolism disorder [33]. According to a recent report, the human body may be exposed to microplastics via two major routes, inhalation and ingestion [34]. Inhalation exposure to airborne microplastics, in particular, can result in particle accumulation by infiltrating deep into the lung [10]. It has been previously reported that human inhalation doses of microplastic ranged from 6.5–8.97 μg/kg/day, with infants receiving doses 3 to 50 times higher than adults [35]. The 5 mg/kg dose of microplastics is approximately 12-fold the daily exposure dose for adults and 600-fold the dose for infants. However, we observed that pulmonary toxicity responses of all three mouse strains administered with 5 mg/kg PVC microplastics, including cellular inflammatory changes in BALF and inflammatory cell infiltration in lung tissue, were not significantly different from those of various control groups. The findings suggest that PVC-induced lung injury in humans is generally as a result of continuous long-term inhalation exposure, raising the need for further long-term studies in the inhalation toxicity of this PVC microplastic.

In the current study, inflammatory cell counts were significantly increased in the BALF of PS-instilled C57BL/6 and ICR mice compared to those in the control groups but not in PS-instilled BALB/c mice (Figure 1). In particular, the eosinophils and neutrophils in the BALF of PS-stimulated C57BL/6 mice were increased (Figure 1A). The activation of eosinophils and neutrophils in the respiratory system causes lung inflammation owing to the release of various cytokines and chemokines [36,37,38]. In particular, cytokines released by helper T (Th) cells by exposure to various chemicals and pathogens contribute to the inflammatory response in pulmonary diseases [39,40]. Previous studies have reported that, contrary to C57BL/6 mice showing a Th1-dominant immune response, BALB/c mice showed a Th2-dominant immune response [41]. Low-dose ovalbumin (OVA)-stimulated C57BL/6 mice also showed a Th2-dominant immune response compared with BALB/c mice, but high-dose OVA stimulation showed a Th2-dominant immune response in BALB/c mice [42]. Recent studies have reported that C57BL/6 mice, irrespective of Th1/Th2 responses, are more sensitive to pulmonary eosinophils than BALB/c mice [43]. In addition, we observed that the regulation of eosinophil expression-associated genes including C-X-C motif chemokine ligand 5 (Cxcl5), C-C motif chemokine ligand (CCL)7, and CCL8 was increased in the lung tissue of PS-instilled C57BL/6 mice (Appendix A). These results suggest that PS microplastic stimulation may contribute to eosinophilic lung inflammation.

Studies have shown several cases of long-term PP exposure resulting in interstitial lung disease [44,45]. In addition, PP microplastic inhalation in humans increased impairment in lung functioning and inflammatory cytokine levels such as IL-8 and tumor necrosis factor-alpha (TNF-α) [11]. Previous studies have also reported an increase in cytotoxicity of PP-exposed Raw 264.7 cells through increased ROS production. More importantly, this observation occurred in a size- and concentration-dependent manner, smaller particle sizes triggered a higher inflammation [46]. In the current study, we investigated the pulmonary toxicity of PP microplastics in three mouse strains. Inflammatory cells, such as total cells, macrophages, and neutrophils, were significantly increased in the BALF of PP-instilled ICR mice but not in C57BL/6 and BALB/c mice (Figure 1). Our results show that inflammatory cell infiltration in lung tissue of PP-stimulated ICR mice increased compared to the control (Figure 2C, Table 3). Inflammatory cells including macrophages and neutrophils are known to play important roles in the early stages of inflammation [47]. Macrophages function mainly to phagocytize bacteria, damaged tissue, and particulate matter, and is one of the main mechanisms of the innate immune system defense [48]. Previous studies have reported that polymer microspheres can be phagocytosed by macrophages [49]. In addition, PP microplastic exposure might cause health deterioration by cytokine releases by immune cells including macrophages [46]. Thus, the results of the present study also suggest that PP stimulation can contribute to the pathogenesis of lung inflammation by the recruitment of immune cells.

The NLRP3 inflammasome is activated through various signals and stimuli, such as particulate matter and environmental pollutants, leading to the release of inflammatory cytokines, such as IL-1β and IL-18 [19,22,50,51]. IL-1β release mediates the release of other inflammatory cytokines, including TNF-α and IL-6, which play essential roles in the pathogenesis of pulmonary diseases such as asthma, pulmonary fibrosis, and COPD [29,52]. Recent studies have reported that Caspase-1 and NLRP3 levels in idiopathic pulmonary fibrosis and asthmatic patients are higher than those in healthy patients [53,54]. Previous studies have shown that activation of NLRP3 inflammasome is observed in HDM and fungi-induced asthma, and NLRP3 inflammasome inhibition mitigates asthmatic symptoms such as Th2 cytokine release, airway hyperresponse, and inflammatory responses [28,55]. Furthermore, recent studies have shown that BLM-induced pulmonary fibrosis increases protein levels of NLRP3 inflammasome components, including NLRP3, ASC, and Caspase-1, whereas NLRP3 deficiency results in recovery of lung fibrosis [56]. Our results showed that NLRP3 inflammasome components such as NLRP3, ASC, and Caspase-1 in lung tissue of PS-instilled C57BL/6 mice and PP-instilled ICR mice were increased. IL-1β levels also increased significantly compared to control groups (Figure 3 and Figure 4). These results suggest that PS and PP microplastic stimulation may be involved in lung inflammation via the NLRP3 inflammasome signaling pathway.

## 4. Materials and Methods

### 4.1. Animals and Experimental Design

Seven-week-old male C57BL/6, BALB/c, and ICR mice were purchased from Orient Bio, Inc. (Seongnam, Korea). The mice were housed in a temperature-controlled environment (22 ± 3 °C) with a relative humidity of 50 ± 20%, a 12 h light/dark cycle, and ventilated with air (10–20 times/h). The mice were provided with pellets that were specifically produced for experimental animals (PMI Nutrition International, Richmond, IN, USA) and UV-irradiated (Steritron SX-1; Daeyoung, Seoul, Korea) and filtered (1 μm-pore filter) tap water. All experimental procedures were approved by the Institutional Animal Care and Use Committee of the Korea Institute of Toxicology (IACUC #2108-0023). The mice in the PS, PP, and PVC groups received intratracheal instillation of 5 mg/kg PS, PP, or PVC in 50 μL of saline solution for two weeks using an automatic video instillator. Mice in the VC group were instilled with saline using the same method. The mice were sacrificed on day 15.

### 4.2. BALF Preparation

At 24 h after the last microplastic intratracheal instillation, the mice were anesthetized with isoflurane and exsanguinated. The left lung was ligated, and the right lung was gently lavaged three times via a tracheal tube with a total volume of 0.7 mL of phosphate-buffered saline (Gibco, Grand Island, NY, USA). The collected solutions were pooled and stored at 4 °C. BAL cells were prepared using Cytospin (Thermo Fisher Scientific, Inc., Waltham, MA, USA) and stained with Diff-Quik solution (Dade Diagnostics, Aguada, Puerto, USA). In total, 200 cells were counted on each slide.

### 4.3. Measurement of Inflammatory Cytokine and Chemokine Levels in BALF 

IL-1β and IL-6 levels in BALF were quantified by ELISA using a commercial kit (R&D Systems, Inc. Minneapolis, MN, USA) in accordance with the manufacturer’s protocols.

### 4.4. Histopathological Analysis

The left lung of each mouse was fixed in 10% neutral-buffered formalin (NRF). The specimens were dehydrated and embedded in paraffin to produce tissue blocks, which were sectioned into 4-µm thick slices. Lung sections from each animal were stained with hematoxylin and eosin (H&E). All samples were analyzed using a Leica DM2500 microscope (Leica Instruments, Wetzlar, Germany) at 200× magnification. The degree of lung injury in each animal was scored on a scale of 0–4, as follows: 0: no symptoms; 1: minimal; 2: slight; 3: moderate; 4: severe.

### 4.5. Preparation of Cell Lysates and Western Blot Analysis

Cell lysates were homogenized in the presence of a protease inhibitor cocktail in a radioimmunoprecipitation assay (RIPA) buffer (Thermo Fisher Scientific, Inc., Waltham, MA, USA). Protein concentrations were determined using the Bradford reagent (Bio-Rad Laboratories, Hercules, CA, USA). The samples were then loaded onto sodium dodecyl sulfate-polyacrylamide gel electrophoresis gels. After electrophoresis at 90 V for 120 min, the proteins were transferred to polyvinylidene difluoride membranes (Merck Millipore, Darmstadt, Germany) at 250 mA for 60 min using a transfer method. Nonspecific sites were blocked with 5% nonfat dry milk in Tris-buffered saline/Tween 20 (TBS-T) for 1 h and incubated with antibodies specific for NLRP3 (AdipoGen Life Sciences, Inc. Liestal, Switzerland), ASC (AdipoGen Life Sciences, Inc. Liestal, Switzerland), Caspase-1 (AdipoGen Life Sciences, Inc. Liestal, Switzerland), and β-actin (Santa Cruz Biotechnology, Dallas, TX, USA) overnight at 4 °C. Anti-rabbit (Cell Signaling Technology, Beverly, MA, USA) and anti-mouse (Cell Signaling Technology, Beverly, MA, USA) horseradish peroxidase-conjugated IgG antibodies were used to detect antibody binding. The binding of specific antibodies was visualized using the iBright CL 1000 imaging system (Thermo Fisher Scientific, Inc., Waltham, MA, USA) after treatment with the ECL reagent (Thermo Fisher Scientific, Inc., Waltham, MA, USA). The results of the densitometric analysis were expressed as the relative ratio of the target protein to the reference protein. The ratio of the target protein to the control was arbitrarily denoted as 1.

### 4.6. Statistical Analysis 

Statistical analysis was performed using GraphPad InStat v. 3.0 (GraphPad Software, Inc., La Jolla, CA, USA). Statistical comparisons were performed using one-way ANOVA followed by Dunnett’s multiple comparison test, and statistical comparisons between two groups were performed using a Student’s *t*-test. Data are presented as mean ± SD. A value of *p* < 0.05 was considered statistically significant.

## 5. Conclusions

We investigated the pulmonary toxicity responses to three microplastics (PS, PP, and PVC) in three strains of C57BL/6, BALB/c, and ICR mice. PS-instilled C57BL/6 and PP-instilled ICR mice showed lung inflammatory responses, including inflammatory cell changes and histopathological analysis. However, PVC microplastic exposure had no toxic effects on the three strains of mice. Interestingly, PS and PP microplastic stimulation significantly increased the protein levels of the NLRP3 inflammasome components. These results suggest that PS and PP microplastic stimulation may lead to pulmonary inflammation through the NLRP3 inflammasome signaling pathway.

## Figures and Tables

**Figure 1 molecules-27-07926-f001:**
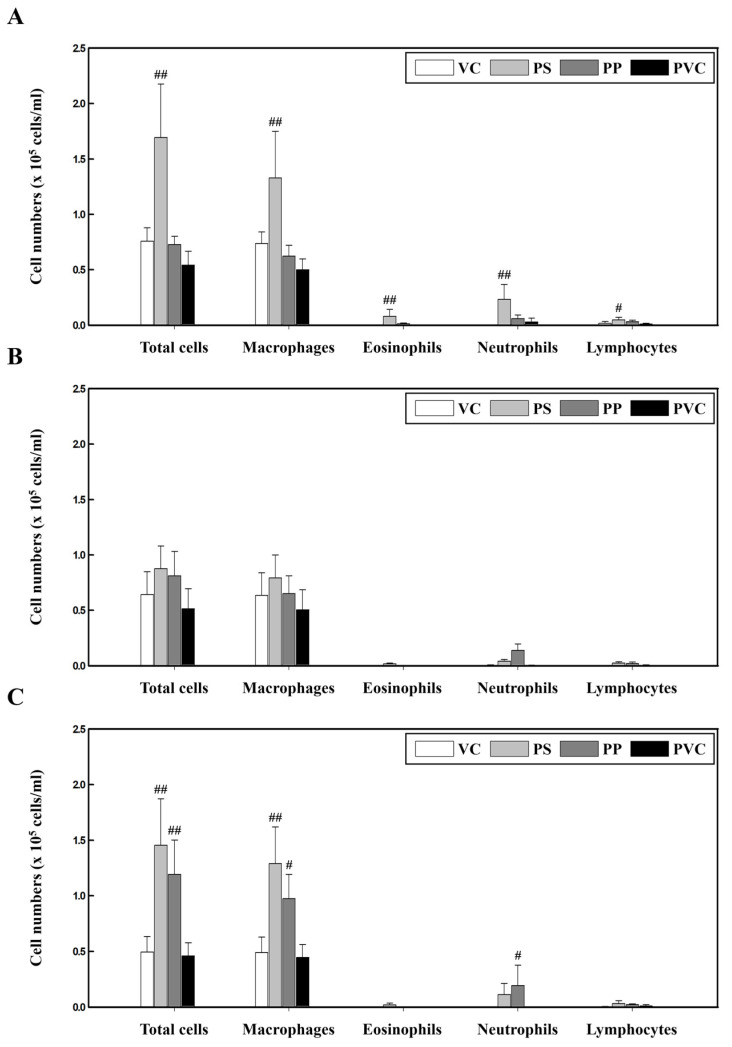
Cellular changes in the BALF of three strains of mice stimulated with three microplastics (PS, PP, and PVC): (**A**) C57BL/6, (**B**) BALB/c, and (**C**) ICR mice. Data are presented as mean ± SD (n = 5 per group). # *p* ≤ 0.05; ## *p* ≤ 0.01 vs. VC.

**Figure 2 molecules-27-07926-f002:**
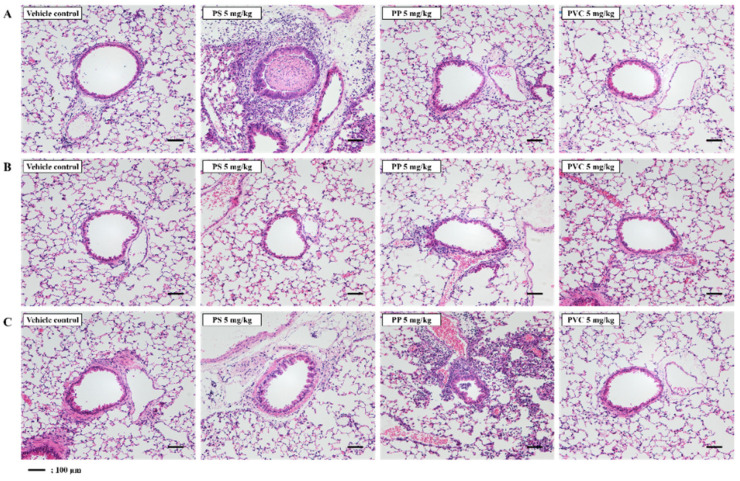
Representative H&E-stained lung tissue section of three strains of mice stimulated with three microplastics (PS, PP, and PVC); (**A**) C57BL/6, (**B**) BALB/c, and (**C**) ICR mice. Scale bar: 100 μm.

**Figure 3 molecules-27-07926-f003:**
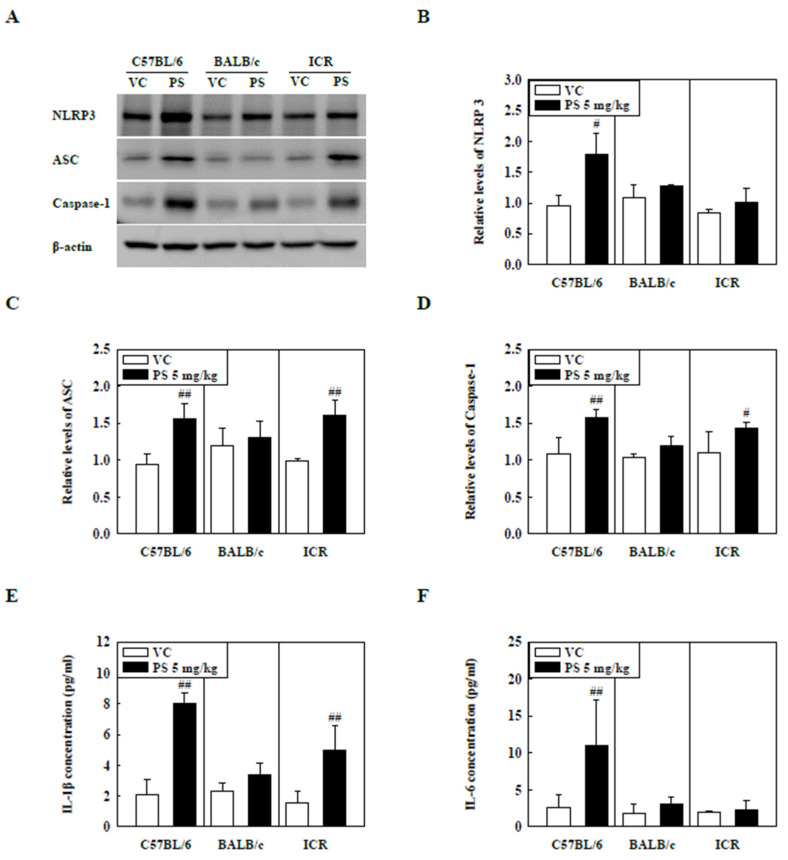
(**A**) Representative Western blot images of NLRP3, ASC, and Caspase-1 expression in lung tissue of three strains of PS-instilled mice. (**B**) Relative density analysis of NLRP3 levels. (**C**) Relative density analysis of ASC levels. (**D**) Relative density analysis of Caspase-1 levels. Data were normalized against β-actin. Inflammatory cytokine levels, including (**E**) IL-1β and (**F**) IL-6, in the BALF of three strains of PS-instilled mice. Data are represented as mean ± SD (n = 5 per group). # *p* ≤ 0.05; ## *p* ≤ 0.01 vs. VC.

**Figure 4 molecules-27-07926-f004:**
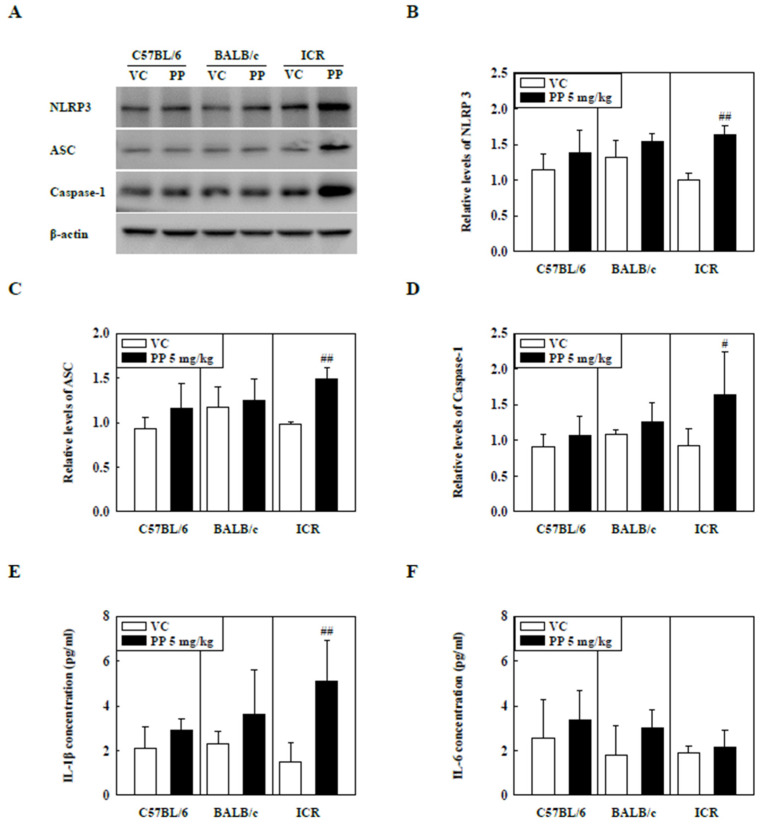
(**A**) Representative Western blot images of NLRP3, ASC, and Caspase-1 expression in lung tissue of three strains of PP-instilled mice. (**B**) Relative density analysis of NLRP3 levels. (**C**) Relative density analysis of ASC levels. (**D**) Relative density analysis of Caspase-1 levels. Data were normalized against β-actin. Inflammatory cytokine levels, including (**E**) IL-1β and (**F**) IL-6, in the BALF of three strains of PP-instilled mice. Data are represented as mean ± SD (n = 5 per group). # *p* ≤ 0.05; ## *p* ≤ 0.01 vs. VC.

**Table 1 molecules-27-07926-t001:** Histopathological findings in the lung of C57BL/6 mice following microplastic intratracheal instillation.

Group	VC	PS (5 mg/kg)	PP (5 mg/kg)	PVC (5 mg/kg)
Number of animals	5	5	5	5
Inflammatory cells infiltration	(0)	(5)	(1)	(2)
Minimal	0	0	0	1
Slight	0	3	1	1
Moderate	0	2	0	0
Mean ± SD	0	2.40 ± 0.55 **	0.40 ± 0.89	0.60 ± 0.89

0, no symptoms; 1, minimal; 2, slight; 3, moderate. Data are presented as the mean ± SD. microplastic group vs. VC group: ** *p* < 0.01.

**Table 2 molecules-27-07926-t002:** Histopathological findings in the lung of BALB/c mice following microplastic intratracheal instillation.

Group	VC	PS (5 mg/kg)	PP (5 mg/kg)	PVC (5 mg/kg)
Number of animals	5	5	5	5
Inflammatory cells infiltration	(0)	(3)	(3)	(0)
Minimal	0	3	3	0
Mean ± SD	0	0.60 ± 0.55	0.20 ± 0.45	0

0, no symptoms; 1, minimal. Data are presented as the mean ± SD.

**Table 3 molecules-27-07926-t003:** Histopathological findings in the lung of ICR mice following microplastic intratracheal instillation.

Group	VC	PS (5 mg/kg)	PP (5 mg/kg)	PVC (5 mg/kg)
Number of animals	5	5	5	5
Inflammatory cells infiltration	(0)	(3)	(5)	(1)
Minimal	0	2	2	1
Slight	0	1	2	0
Mean ± SD	0	0.80 ± 0.84	1.20 ± 0.84 *	0.20 ± 0.45

0, no symptoms; 1, minimal; 2, slight symptoms. Data are presented as the mean ± SD. microplastic group vs. VC group: * *p* < 0.05.

## Data Availability

Not applicable.

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
