# Peer review of "Pulmonary Toxicity of Polystyrene, Polypropylene, and Polyvinyl Chloride Microplastics in Mice"

_molecules, 2022, doi:10.3390/molecules27227926_

Round 1

Reviewer 1 Report

The manuscript presented for review (Pulmonary toxicity of polystyrene, polypropylene, and polyvinyl chloride microplastics in mice) concerns pulmonary toxicity associated with microplastic exposure. The study was conducted on mice susceptible to induced colorectal cancer, resistant to tumour formation, and prone to primary lung tumours. The study was planned correctly, taking into account the well-being of the animals used for the experiment. 

The topic presented in the manuscript is extremely important, given the global, increasing microplastic pollution of the environment including the air. The effect of microplastic on the protein expression of NLRP3, associated with the NF-kappaB signalling pathway, involved in the regulation of inflammation, the immune response, and apoptosis, is described. 

Notes below.

1 The authors indicate that 'The degree of lung injury in each animal was scored on a scale of 0-4. '. What parameters were taken into account, in determining such a scale? Can they be described in the Materials and Methods section, where the scale is mentioned?

2. Readability of Figures 3 and 4 could be slightly improved.

3. in discussion: 'In addition, we observed that the regulation of eosinophil expression-associated genes including C-X-C motif chemokine ligand 5 (Cxcl5), C-C motif chemokine ligand (CCL)7, and CCL8 were increased in the lung tissue of PS-instilled C57BL/6 mice (data not shown).' why not shown? This is quite interesting. 

The manuscript is well prepared, and the minor comments do not detract from it. 

Author Response

1. The authors indicate that 'The degree of lung injury in each animal was scored on a scale of 0-4. '. What parameters were taken into account, in determining such a scale? Can they be described in the Materials and Methods section, where the scale is mentioned?

Response: Thank you for your valuable comments. We have added the following information to the manuscript according to your comments.

The degree of lung injury in each animal was scored on a scale of 0–4, as follows: 0: no symptoms; 1: minimal; 2: slight; 3: moderate; 4: severe.

We have modified the sentence in lines 302-304.

1). Schafer, K.A.; Eighmy, J.; Fikes, J.D.; Halpern, W.G.; Hukkanen, R.R.; Long, G.G.; Meseck, E.M.; Patrick, D.J.; Thibodeau, M.S.; Wood, C.E.; et al. Use of severity grades to characterize histopathologic changes. Toxicol Pathol. 2018, 46, 256-265.

2. Readability of Figures 3 and 4 could be slightly improved.

Response: Thank you for your valuable comments. We modified Figures 3 and 4 by visually increasing the size of the picture as suggested, to a more readable form.

3. in discussion: 'In addition, we observed that the regulation of eosinophil expression-associated genes including C-X-C motif chemokine ligand 5 (Cxcl5), C-C motif chemokine ligand (CCL)7, and CCL8 were increased in the lung tissue of PS-instilled C57BL/6 mice (data not shown).' why not shown? This is quite interesting.

Response: Thank you for your valuable comments. We have modified the manuscripts in accordance with your comments as follows. We have also added Supplementary Table 1 and Supplementary methods.

We have modified the sentence in line 230-231.

Supplementary Methods

1.1 RNA isolation, library preparation, and sequencing

Total RNA was isolated using Trizol reagent (Invitrogen). RNA quality was assessed using Agilent 2100 bioanalyzer with the RNA 6000 Nano Chip (Agilent Technologies, Amstelveen, Netherlands), and RNA quantification was performed using a ND-2000 Spectrophotometer (Thermo Fisher Scientific). Only samples with an A260/A280 ratio >1.8 and RIN value >7 were considered suitable for use. For control and test RNAs, a library was constructed using QuantSeq 3′ mRNA-Seq Library Prep Kit (Lexogen, Inc., Austria) according to the manufacturer’s instructions. Each 500-ng sample of total RNA was prepared and an oligo-dT primer containing an Illumina-compatible sequence at its 5′ end was hybridized, and reverse transcription performed. After degradation of the RNA template, second-strand synthesis was initiated by a random primer containing an Illuminacompatible linker sequence at its 5′ end. The double-stranded library was purified by using magnetic beads to remove all reaction components. The library was amplified to add the complete adapter sequences required for cluster generation. The finished library was purified from PCR components. High-throughput sequencing was performed through single-end 75 sequencing using NextSeq 500 (Illumina, Inc., USA).  

1.2 Data analysis

QuantSeq 3′ mRNA-Seq reads were aligned using Bowtie2. Bowtie2 indices were either generated from genome assembly sequences or the representative transcript sequences for aligning to the genome or transcriptome. The alignment file was used for assembling transcripts, estimating their abundances, and detecting differential expression of genes. Differentially expressed genes (DEGs) were determined based on counts from unique and multiple alignments using coverage in Bedtools (Quinlan AR, 2010). Read count (RC) data were processed based on the quantile normalization method using EdgeR within R and Bioconductor. Gene classification was based on searches in the DAVID (http://david.abcc.ncifcrf.gov/).

1.3 Gene ontology (GO) category analysis   

To classify the genes altered in PS-instilled C57BL/6 mice with a similar pattern of expression, each gene was assigned to an appropriate category according to its main cellular function. To determine significantly over-represented GO findings, the DAVID functional annotation clustering tool was used by choosing the default option. A Fisher exact test was used to identify significantly enriched pathways and the resulting P values were adjusted using the BH false discovery rate (FDR) algorithm. Pathway categories with FDR < 0.05 were reported.

Supplementary Table 1. Genes altered in the regulation of eosinophil expression

Symbol

Entrez Gene Name

Fold change

p-value

CXCL5

chemokine (C-X-C motif) ligand 5

5.321

0.000

CCL7

chemokine (C-C motif) ligand 7

2.183

0.041

CCL8

chemokine (C-X-C motif) ligand 8

6.985

0.004

Reviewer 2 Report

In this work, the authors use three mouse strains (C57BL/6, BALB/c, and ICR) to evaluate the effects of polystyrene, polypropylene, and polyvinyl chloride following intrathecal administration, using an automatic video instillator, for 14 days.

The bronchoalveolar lavage fluid of the instilled mice was collected and the cell type and number, as well as the levels of IL-1β and IL-6 were assessed. Moreover, lung tissue sections were H&E stained to assess pathological alterations and the levels of NLRP3, ASC and Caspase-1 were estimated by Western Blotting.

The manuscript is well-written, with the introduction providing the required background information, the results section providing the findings in a clear and efficient way, and the materials and methods describing the experimental procedures in an adequate level of detail. There is a logical continuum in the presentation of the results and the conclusions are supported by the results.

I think that the manuscript would benefit from a stronger emphasis on the pathological mechanisms triggered by the instillation of the microplastics. For example, what type of immune response (Th1 or Th2) is initiated in the mice? To this end, expression data of Cxcl5, CCL7 and CCL8 should be included in the manuscript, even as supplementary data (line 228). Similarly, the histopathological study could be extended with some immunohistochemical stainings, to better characterize the immune response. Moreover, it is very interesting to see that the three mouse strains react differently to the microplastics and I think that the authors could focus more on the possible explanations of this divergent reaction among the three mouse strains.

Author Response

1. I think that the manuscript would benefit from a stronger emphasis on the pathological mechanisms triggered by the instillation of the microplastics. For example, what type of immune response (Th1 or Th2) is initiated in the mice? To this end, expression data of Cxcl5, CCL7 and CCL8 should be included in the manuscript, even as supplementary data (line 228). Similarly, the histopathological study could be extended with some immunohistochemical stainings, to better characterize the immune response. Moreover, it is very interesting to see that the three mouse strains react differently to the microplastics and I think that the authors could focus more on the possible explanations of this divergent reaction among the three mouse strains.

Response: Thank you for your valuable comments. We observed that three microplastics induced different immune response in three strains of mice. Especially, PS microplastics contributed to the recruitment of eosinophils in C57BL/6 mice. We agree that data for regulation of eosinophil expression-related genes in lung tissue of PS-instilled C57BL/6 mice were needed. Therefore, we have added Supplementary Table 1 and Supplementary methods according to reviewer comments.  

We have modified the sentence in line 230.

Supplementary Methods

1.1 RNA isolation, library preparation, and sequencing

Total RNA was isolated using Trizol reagent (Invitrogen). RNA quality was assessed using Agilent 2100 bioanalyzer with the RNA 6000 Nano Chip (Agilent Technologies, Amstelveen, Netherlands), and RNA quantification was performed using a ND-2000 Spectrophotometer (Thermo Fisher Scientific). Only samples with an A260/A280 ratio >1.8 and RIN value >7 were considered suitable for use. For control and test RNAs, a library was constructed using QuantSeq 3′ mRNA-Seq Library Prep Kit (Lexogen, Inc., Austria) according to the manufacturer’s instructions. Each 500-ng sample of total RNA was prepared and an oligo-dT primer containing an Illumina-compatible sequence at its 5′ end was hybridized, and reverse transcription performed. After degradation of the RNA template, second-strand synthesis was initiated by a random primer containing an Illuminacompatible linker sequence at its 5′ end. The double-stranded library was purified by using magnetic beads to remove all reaction components. The library was amplified to add the complete adapter sequences required for cluster generation. The finished library was purified from PCR components. High-throughput sequencing was performed through single-end 75 sequencing using NextSeq 500 (Illumina, Inc., USA).  

1.2 Data analysis

QuantSeq 3′ mRNA-Seq reads were aligned using Bowtie2. Bowtie2 indices were either generated from genome assembly sequences or the representative transcript sequences for aligning to the genome or transcriptome. The alignment file was used for assembling transcripts, estimating their abundances, and detecting differential expression of genes. Differentially expressed genes (DEGs) were determined based on counts from unique and multiple alignments using coverage in Bedtools (Quinlan AR, 2010). Read count (RC) data were processed based on the quantile normalization method using EdgeR within R and Bioconductor. Gene classification was based on searches in the DAVID (http://david.abcc.ncifcrf.gov/).

1.3 Gene ontology (GO) category analysis  

To classify the genes altered in PS-instilled C57BL/6 mice with a similar pattern of expression, each gene was assigned to an appropriate category according to its main cellular function. To determine significantly over-represented GO findings, the DAVID functional annotation clustering tool was used by choosing the default option. A Fisher exact test was used to identify significantly enriched pathways and the resulting P values were adjusted using the BH false discovery rate (FDR) algorithm. Pathway categories with FDR < 0.05 were reported.

Supplementary Table 1. Genes altered in the regulation of eosinophil expression

Symbol

Entrez Gene Name

Fold change

p-value

CXCL5

chemokine (C-X-C motif) ligand 5

5.321

0.000

CCL7

chemokine (C-C motif) ligand 7

2.183

0.041

CCL8

chemokine (C-X-C motif) ligand 8

6.985

0.004